# Diffusion-based Prompt Generation for Lifelong Continual Adaptation

## Abstract

Continual Test-time Adaptation (TTA) addresses sequential out-of-distribution scenarios with unlabeled data but overlooks long-term and recurring in-distribution aspects of the real world. Therefore, we introduce Lifelong Continual Adaptation, which enables models to efficiently retrieve domain-specific knowledge when encountering in-distribution data streams with sequential and recurring domains. We found that optimization-based Continual TTA methods underperform on the proposed problem due to two major pitfalls: updating the model's parameters is expensive and impractical for resource-constrained devices, and these methods exhibit instability when adapting to long-term recurring domains. To address these challenges, we propose a diffusion-based prompt generation method (DiffPrompt). Specifically, instead of continually optimizing the foundation model, we generate domain-specific prompts for it to adapt. We use a conditional diffusion model to learn a prompt-space distribution for various domains. During testing, the diffusion model generates prompts for the current domain based on the incoming batch of data, facilitating the continual adaptation of the foundation model. Our experiments demonstrate that DiffPrompt enables stable and efficient deployment in practical scenarios involving sequential and recurring domains.

## 1 Introduction

Domain shifts significantly degrade the performance of deep learning models due to the misalignment between the domains of the training and deployment phases (Quionero-Candela et al., 2009; Koh et al., 2021). This challenge is further compounded when models encounter sequential domain changes during deployment (Wang et al., 2022). For example, the driving conditions for autonomous vehicles are always evolving such as lighting conditions at various times of the day and varying weather conditions. It causes different visual data, demanding the model to adapt wisely.

To address these challenges, a line of research known as continual Test-Time Adaptation (TTA) focuses on continually adapting machine learning models. They typically aim to address sequential novel domains with unlabeled data examples encountered during deployment. The assumption is that both data instances and domains are previously unseen during training (Wang et al., 2022; Döbler et al., 2023; Yuan et al., 2023). However, the extant framework of continual TTA fails to account for three critical real-world scenarios. First, they emphasize sequential out-of-distribution (OOD) scenarios, where training and deployment domains differ, while overlooking the sequential in-distribution (ID) data, resulting in an incomprehensive evaluation. Second, the scarcity of the number of sequential domains limits the assessment of a model's long-term adaptation capabilities. Finally, the recurring nature of certain domains encountered repetitively during deployment is disregarded (Hoang et al., 2023). Therefore, we introduce a novel problem setting, known as Lifelong Continual Adaptation (LCA), that aims to foster model continual adaptation in more realistic scenarios characterized by long-term sequential, recurring, and in-distribution data, described in Table 1 and Figure 1. LCA explores how to efficiently retrieve and utilize knowledge of seen domains during deployment.

We conducted an analysis of multiple baseline methods on the proposed LCA setting, including the non-adapted domain generalization approach, Empirical Risk Minimization (ERM) (Vapnik et al., 1998), and various continual TTA methods (Wang et al., 2022; Döbler et al., 2023; Yuan et al., 2023).

Table 1: Distinguishing between Continual TTA and our Lifelong Continual Adaptation setting (LCA). The goal of Continual TTA is to preserve model performance from degrading on sequential domain shifts by learning from unseen domains during test time, while the goal of LCA is to achieve stable and high performance on realistic domain distributions by retrieving knowledge for each encountered seen domain during deployment.

| Setting | ID or OOD test streams | Sequential domains | Long-term and recurring |
|---|---|---|---|
| Continual TTA | OOD | ✓ | ✗ |
| LCA (Ours) | ID | ✓ | ✓ |

Figure 1: This paper considers a scenario where a model operates in data streams featuring sequential and recurring domains. Different colors represent different domains. The Deployment part illustrates the continual adaptation workflow of the proposed prompt generation method at deployment time. The training process is illustrated in Figure 2.

In our findings, ERM achieves only moderate performance due to its lack of domain adaptation when encountering sequential domain shifts. Conversely, continual TTA approaches endeavor to adapt to different sequential domains by optimizing model parameters using manual-crafted self-supervised training objectives. However, these methods present two primary drawbacks. First, the optimization of model parameters demands substantial computational and memory resources, rendering deployment on resource-constrained devices impractical (Bommasani et al., 2021). Even worse, the recent trend of scaling laws in foundation models exacerbates the issue (Kaplan et al., 2020). Second, our empirical results, as presented in Table 4, reveal that optimization-based continual TTA methods exhibit significant instability, impeding their capability to adapt to long-term recurring domains.

Alternatively, as a paradigm of parameter-efficient fine-tuning, prompt tunning (Zhou et al., 2022b; Jia et al., 2022) can guide foundation models to adapt to downstream tasks by optimizing only the learnable prompt vectors prepended to the input space, while maintaining the rest of the model frozen. Inspired by this paradigm, we propose a prompt generation framework that enables vision foundation models, such as CLIP (Radford et al., 2021), to adapt to long-term sequential domains under the resource-constrained scenario. Specifically, we introduce a diffusion model as a domain prompt generator, directly sampling a domain prompt through an iterative denoising process conditioned on incoming batches of data examples from each specific domain. The training of the diffusion-based prompt generator is decomposed into two stages: prompt collection and diffusion training. The primary objective of the prompt collection stage is to encapsulate the prompt distribution of each training domain through a set of learnable prompt samples, wherein domain-specific knowledge is presumed to be encoded within each domain's prompt distribution. During the diffusion training stage, we train a conditional latent diffusion model using the collected prompt samples from each specific domain. This model aims to generate domain-specific prompts from Gaussian noise, conditioned on mini-batch data from that domain. The principal advantage of employing a diffusion-based generative approach to represent domain-specific knowledge lies in its robustness compared to discriminative counterparts, as it learns the distribution of domain prompts rather than relying on a single domain prompt. In addition, directly generating a domain prompt without prompt tuning alleviates gradient backpropagation, leading to significant resource efficiency during adaptation.

Our work makes three contributions. First, we introduce a novel problem setting for continual domain adaptation, known as Lifelong Continual Adaptation, which emphasizes the evaluation of long-term sequential, recurring, and in-distribution domains in real-world scenarios. Second, we propose a framework that formulates the continual adaptation process as generating a sequence of domain-

specific prompts to guide the foundation model for domain adaptation, utilizing a diffusion-based generative approach. Diffusion models have shown success in generating inputs (e.g. images (He et al., 2023)), outputs (e.g. bounding boxes (Chen et al., 2023), semantic labels (Tan et al., 2022)), and neural network parameters (Erkoç et al., 2023). Our work is the first to show that diffusion models can also be used to generate prompts in continual adaptation. Finally, we conduct extensive experiments to demonstrate the superiority of the proposed method compared with various baselines.

## 2 RELATED WORK

**Prompt tuning and adaptation**  Prompting (Petroni et al., 2019; Brown et al., 2020; Lester et al., 2021; Li & Liang, 2021; Liu et al., 2021; Yao et al., 2023; Zhu et al., 2023) has emerged as a crucial technique for enhancing the performance of pre-trained models in various downstream tasks. Radford et al. (2021) introduces CLIP, a powerful vision-language model which uses textual prompts to guide image classification. Following this, CoOp (Zhou et al., 2022b) proposes to adapt CLIP by learning textual prompts on the text encoder. CoCoOp (Zhou et al., 2022a) extends CoOp by conditionally tuning to improve performance. VPT (Jia et al., 2022) introduces fine-tuning prompts on Vision Transformers (Dosovitskiy et al., 2021) to adapt to downstream tasks.

**Continual adaptation**  Some work (Hoffman et al., 2014; Wulfmeier et al., 2018; Volpi et al., 2021; Liu et al., 2020; Kumar et al., 2020) considers an evolving domain adaptation where the target domain evolves over time. A line of recent research known as continual Test-Time Adaptation (TTA) focuses on continually adapting a source model to target unseen sequential domains (Wang et al., 2022). These methods are mainly based on a teacher-student self-training framework and utilize source prototype pulling (Döbler et al., 2023) and resampling (Yuan et al., 2023) strategies to improve stability. The LCA setting in our work differs from continual TTA in its focus. Continual TTA aims to prevent performance degradation across sequential domain shifts by learning from unseen domains during test time. In contrast, LCA focuses on achieving stable and high performance by retrieving knowledge for each encountered seen domain during deployment.

**Continual learning**  Continual learning addresses catastrophic forgetting, the performance degradation on old tasks when learning new ones (Wang et al., 2024b). Domain incremental learning, a subset of continual learning, involves sequential domains and aims to balance performance across old and new domains (Mirza et al., 2022; Shi & Wang, 2023). In contrast, LCA focuses on retrieving and utilizing knowledge of an old domain during test time, without learning new domains.

**Diffusion-based generation**  Denoising Diffusion Probabilistic Models (DDPM) have garnered significant attention for their ability to produce high-quality data through a process of iterative denoising (Ho et al., 2020; Luo, 2022; Rombach et al., 2022; Dhariwal & Nichol, 2021; Peebles & Xie, 2023; Croitoru et al., 2023). Several studies employ diffusion models for data augmentation (He et al., 2023; Trabucco et al., 2024), and these models are also explored for classification tasks (Li et al., 2023; Du et al., 2023; Prabhudesai et al., 2023). Furthermore, recent research investigates the application of diffusion models for generating neural network weights (Erkoç et al., 2023; Nava et al., 2023; Wang et al., 2024a). Some work also utilizes diffusion models to generate bounding boxes for object detection (Chen et al., 2023) and to enhance the quality of semantic segmentation (Tan et al., 2022).

## 3 METHODOLOGY

We propose a diffusion-based prompt generation method to achieve stable and high-performance deployment of a foundation model on practical data streams involving sequential and recurring domains. Section 3.1 introduces the preliminaries of the problem definition and diffusion models. Section 3.2 describes the training of our method prior to deployment. Section 3.3 introduces the condition module within our method. Finally, Section 3.4 covers the deployment of our method.

### 3.1 PRELIMINARY

#### 3.1.1 PROBLEM DEFINITION

Let $\mathcal{D}_1, \mathcal{D}_2, \ldots, \mathcal{D}_n$ represent different domains, each with a corresponding training set $\mathcal{D}_i^{\text{train}}$ and a test set $\mathcal{D}_i^{\text{test}}$. During **training**, the model has access to the training sets $\{\mathcal{D}_i^{\text{train}}\}_{i=1}^n$ from these

multiple domains. During **deployment**, the model encounters test sets from these domains in a sequential and recurring manner. Let $\mathcal{S} = \{\mathcal{D}_1, \mathcal{D}_2, \ldots, \mathcal{D}_n\}$ represent the sequence of domains. The test stream presents this sequence of domains, which recurs $r$ times, as represented:

$$S^{test} = \{(D_1^{test}, D_2^{test}, \ldots, D_n^{test})_1, (D_1^{test}, D_2^{test}, \ldots, D_n^{test})_2, \ldots, (D_1^{test}, D_2^{test}, \ldots, D_n^{test})_r\} \tag{1}$$

Specifically, the model performs adaptation and inference on the data examples from each domain $D_i^{\text{test}}$. We use the classification accuracy $A_{i,j}$ as the evaluation metric, corresponding to the performance on the $i$-th domain during the $j$-th recurrence. In addition, we calculate the mean accuracy over the entire data stream as follow:

$$\bar{A} = \frac{1}{nr} \sum_{j=1}^{r} \sum_{i=1}^{n} A_{i,j}. \tag{2}$$

### 3.1.2 Diffusion models

Diffusion models are a sophisticated class of generative models that have shown remarkable capabilities in generating high-quality synthetic data. The core principle behind diffusion models involves a process known as the forward and reverse diffusion processes (Ho et al., 2020; Luo, 2022; Rombach et al., 2022).

**Forward diffusion**   The original data is gradually noised in forward diffusion. Specifically, for data $\boldsymbol{x}_0$ sampled from the real distribution $q(\boldsymbol{x})$, the forward diffusion $q(\boldsymbol{x}_{1:T}|\boldsymbol{x}_0)$ is a process of adding noise to the data with a Markov chain of $T$ steps of $q(\boldsymbol{x}_t|\boldsymbol{x}_{t-1})$, at each of which Gaussian noise with variance $\beta_t$ is added:

$$q(\boldsymbol{x}_{1:T}|\boldsymbol{x}_0) = \prod_{t=1}^{T} q(\boldsymbol{x}_t|\boldsymbol{x}_{t-1}), \text{ where } q(\boldsymbol{x}_t|\boldsymbol{x}_{t-1}) = \mathcal{N}(\boldsymbol{x}_t; \boldsymbol{\mu}_t = \sqrt{1-\beta_t}\boldsymbol{x}_{t-1}, \boldsymbol{\Sigma}_t = \beta_t\boldsymbol{I}), \tag{3}$$

where $\boldsymbol{\mu}$ and $\boldsymbol{\Sigma}$ are the mean and variance, and $\boldsymbol{I}$ is the identity matrix.

**Reverse diffusion**   The forward process adds noise incrementally until $\boldsymbol{x}_T$ resembles isotropic Gaussian noise. Consequently, we can sample a $\boldsymbol{x}_T$ from a Gaussian distribution $\mathcal{N}(0, \boldsymbol{I})$ and conduct a reverse diffusion to generate a sample $\boldsymbol{x} \sim q(\boldsymbol{x})$. Because $q(\boldsymbol{x}_{t-1}|\boldsymbol{x}_t)$ is intractable to compute, we use a neural network $p_\theta$ parameterized with $\theta$ to approximate it:

$$p_\theta(\boldsymbol{x}_{0:T}) = p_\theta(\boldsymbol{x}_T) \prod_{t=1}^{T} p_\theta(\boldsymbol{x}_{t-1}|\boldsymbol{x}_t), \text{ where } p_\theta(\boldsymbol{x}_{t-1}|\boldsymbol{x}_t) = \mathcal{N}(\boldsymbol{x}_{t-1}; \boldsymbol{\mu}_\theta(\boldsymbol{x}_t, t), \boldsymbol{\Sigma}_\theta(\boldsymbol{x}_t, t)), \tag{4}$$

where $\boldsymbol{\mu}_\theta$ and $\boldsymbol{\Sigma}_\theta$ are the predicted Gaussian parameters by the diffusion model.

The training of the diffusion model involves the optimization of the negative log-likelihood of the training data. A simplified version of the evidence lower bound is typically used as the objective function:

$$\mathcal{L} = \mathbb{E}_{\boldsymbol{x}_0, t, \boldsymbol{\epsilon} \sim \mathcal{N}(0, \boldsymbol{I})} \left[ \|\boldsymbol{\epsilon} - \boldsymbol{\epsilon}_\theta(\sqrt{\bar{\alpha}_t}\boldsymbol{x}_0 + \sqrt{1-\bar{\alpha}_t}\boldsymbol{\epsilon}, t)\|^2 \right], \tag{5}$$

where $\alpha_t = 1 - \beta_t$ and $\bar{\alpha}_t = \prod_{s=1}^{t} \alpha_s$; $\boldsymbol{\epsilon}_\theta$ is the neural network used to predict the noise $\boldsymbol{\epsilon}$ at each time step $t$.

### 3.2 Overview of training

There are two stages during the training of our method: prompt collection and diffusion training, as illustrated in Figure 2. In the prompt collection stage, we train a base model with trainable prompts using the training sets of different domains to collect prompt samples. In the diffusion training stage, we train a diffusion model with the collected prompt samples for prompt generation at deployment time.

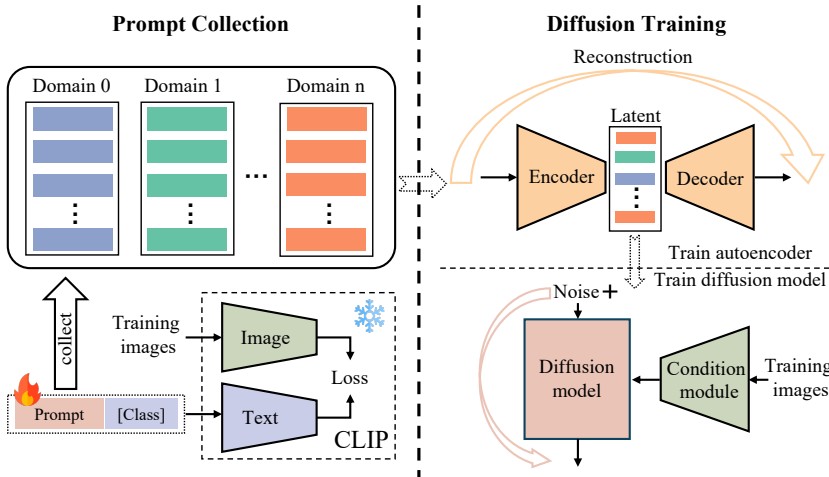

Figure 2: Overview of training. The training of our method involves two stages: prompt collection (left) and diffusion training (right). In the prompt collection stage, a base model with trainable prompts is trained using the training sets from different domains to collect prompt samples. In the diffusion training stage, these collected prompt samples are used to train a conditional latent diffusion model for prompt generation at deployment time. The deployment process is illustrated in Figure 1.

**Prompt collection** We employ the foundation model CLIP (Radford et al., 2021) as the base model. Upon it, we adopt trainable textual prompts (Zhou et al., 2022b) to realize adaptation for the model. On each training set of domains $\{\mathcal{D}_i^{\text{train}}\}_{i=1}^n$, we train the base model using the cross-entropy loss function:

$$\mathcal{L}_{\text{CE}} = -\log \frac{\exp(\cos(\boldsymbol{F}, \boldsymbol{T}_y)/\tau)}{\sum_{j=1}^C \exp(\cos(\boldsymbol{F}, \boldsymbol{T}_j)/\tau)}, \tag{6}$$

where $\boldsymbol{F}$ is the image encoder output; $\boldsymbol{T}_j$ is the text encoder output for the $j$-th class out of $C$ classes; $\cos(\cdot, \cdot)$ denotes the cosine similarity, and $\tau$ is a temperature parameter. Only the prompt is tunable while the whole CLIP model is frozen. For each domain, we collect a set of fitted prompts from different epochs, expressed as:

$$\mathcal{P} = \{\mathcal{P}_k \mid k = 1, 2, \ldots, n\}, \quad \text{where} \quad \mathcal{P}_k = \{\boldsymbol{p}_{k,j} \mid j = 1, 2, \ldots, m\}, \tag{7}$$

where $m$ is the number of prompt samples for each domain.

**Diffusion training** After collecting the prompt samples, we train a conditional latent diffusion model with them to learn the prompt-space distribution among different domains. We first train an unconditional denoising autoencoder $f_{ae}$ to translate the prompts $\mathcal{P}$ to a low-dimensional latent space. It is optimized using a reconstruction loss:

$$\mathcal{L}_{ae} = \frac{1}{B} \sum_{i=1}^B \|\boldsymbol{p}_i - f_{ae}(\boldsymbol{p}_i, \boldsymbol{\xi})\|^2 \tag{8}$$

where $\boldsymbol{p}_i$ represents the $i$-th prompt in a batch; $\boldsymbol{\xi}$ is the random noise added to the input and the latent space; $f_{ae}(\boldsymbol{p}_i, \boldsymbol{\xi})$ is the reconstructed prompt from the autoencoder with added noise, and $B$ is the batch size. The latent space aids in efficient and stable training (Rombach et al., 2022).

Under the latent space, we train an image-conditioned diffusion model using the prompt samples and training images. This process can be expressed as:

$$\theta = \theta - \gamma \nabla_\theta \left\| \boldsymbol{\epsilon} - \boldsymbol{\epsilon}_\theta \left( \sqrt{\bar{\alpha}_t} \boldsymbol{v}_0 + \sqrt{1 - \bar{\alpha}_t} \boldsymbol{\epsilon}, t, \mathcal{C}(\boldsymbol{X}^{\text{train}}) \right) \right\|^2. \tag{9}$$

Here, $\boldsymbol{v}_0$ is the latent representation of the prompts, and $\mathcal{C}$ is the designed condition module used to encode images into conditions, which will be described in Section 3.3. $\boldsymbol{X}^{\text{train}}$ represents the training images from the corresponding domain associated with $\boldsymbol{v}_0$. $\gamma$ is the learning rate.

### 3.3 Feature distribution as condition

The condition module is designed to perceive domains in order to provide conditions that are sensitive to different domains. Inspired by discussions on the relationship between domains and feature distribution in the field of domain adaptation, we recognize that domain shifts result in varied feature distributions extracted by discriminative models. A line of research focuses on overcoming this issue by forcing models to extract similar feature distributions from images of different domains to achieve domain adaptation (Ganin et al., 2016; Ganin & Lempitsky, 2015; Sun & Saenko, 2016). In this work, however, we exploit this characteristic within the condition module; in other words, we use feature distributions as conditions.

Concretely, we employ a pre-trained image encoder from CLIP in the condition module. It produces features $\boldsymbol{F}$ from the incoming batch of images from streams. Afterward, we compute the distribution statistics for the batch of features, adopting the mean and standard deviation as follows:

$$\boldsymbol{\mu} = \frac{1}{c}\sum_{i=1}^{c}\boldsymbol{F}_i, \quad \boldsymbol{\sigma} = \sqrt{\frac{1}{c}\sum_{i=1}^{c}(\boldsymbol{F}_i - \boldsymbol{\mu})^2}, \tag{10}$$

where $c$ is the batch size of images. We concatenate the mean and standard deviation as $\text{concat}[\boldsymbol{\mu}, \boldsymbol{\sigma}]$ to serve as the conditions to be input to the diffusion model.

### 3.4 Image-conditioned prompt generation

At deployment time, as shown in Figure 1, we sample prompts conditioned on batches of test images from test streams using the trained diffusion model through the reverse diffusion process:

$$\boldsymbol{v}_{t-1} = \frac{1}{\sqrt{\alpha_t}}\left(\boldsymbol{v}_t - \frac{1-\alpha_t}{\sqrt{1-\bar{\alpha}_t}}\boldsymbol{\epsilon}_\theta(\boldsymbol{v}_t, t, \mathcal{C}(\boldsymbol{X}^{\text{test}}))\right) + \sigma_t\boldsymbol{z}, \quad t = T, \ldots, 1, \tag{11}$$

where $\boldsymbol{v}_T$ and $\boldsymbol{z}$ are random noise sampled from $\mathcal{N}(0, \boldsymbol{I})$; $\sigma_t$ comes from the noise schedule used in the forward diffusion process, and $\boldsymbol{X}^{\text{test}}$ represents the test images. The generated prompts are assigned to the base model to adapt it to the current domain.

## 4 Experiments

In this section, we provide a detailed evaluation of our proposed DiffPrompt. We begin by describing the datasets used (4.1) and the experimental setup (4.2). Next, we discuss the baselines against which our method is compared (4.3) and provide implementation details to ensure reproducibility (4.4). Following this, we present the results of our experiments (4.5), demonstrating the efficacy of our approach. Finally, we conduct ablation studies (4.6) to verify whether DiffPrompt functions as intended.

### 4.1 Datasets

We include two datasets in the experiments, DomainNet (Peng et al., 2019) and ImageNet-C (Croce et al., 2021), which are widely used in domain adaptation and test-time adaptation tasks.

**DomainNet** The dataset contains 6 domains: Clipart (clip), Infograph (info), Painting (paint), Quickdraw (quick), Real (real), and Sketch (sketch). It has 345 classes and is naturally class-imbalanced in each domain. We follow the official split to organize the training and test sets for each domain.

**ImageNet-C** This dataset is derived by applying various corruptions to the images in the validation set of ImageNet. There are 4 categories of corruptions (weather, noise, blur, digital) aimed at mimicking a range of natural environmental conditions that may be encountered during deployment. Following the RobustBench benchmark (Croce et al., 2021), we adopt 15 corruption domains, including brightness (bri), frosted glass (gla), JPEG compression (jpe), contrast (con), defocus blur (def), impulse noise (imp), motion blur (mot), snow (sno), zoom blur (zoo), frost (fro), pixelation (pix), gaussian noise (gau), elastic transformation (ela), shot noise (sho), and fog (fog). Similar to the split of DomainNet, we adopt a $70\%/30\%$ split to obtain the training and test sets for each domain.

## 4.2 EXPERIMENTAL SETUP

The experimental setting mimics a practical scenario where a model is deployed in data streams featuring sequential and recurring domains. It has two primary conditions. Firstly, a sequence of domains is presented in streams, which is the same setup as recent continual TTA works (Song et al., 2023; Döbler et al., 2023), with associated experimental results detailed in Tables 2 and 3. Secondly, the sequence of domains in the first condition recurs in streams. We set the number of recurring times to 15, and the associated results are shown in Tables 4 and 5. For all methods, we use the same base model, which is the 'ViT-B/16' version of the CLIP model. All methods follow the paradigm of prompt tuning, where only the prompt is updated while CLIP's model parameters remain frozen. The batch size is uniformly set to 64.

## 4.3 BASELINES

We include 5 baseline methods in the comparative experiments: Zero-shot, Empirical Risk Minimization (ERM), two fully test-time adaptation methods: TENT (Wang et al., 2021) and SAR (Niu et al., 2023), and three recent continual test-time adaptation methods: RMT (Döbler et al., 2023), CoTTA (Wang et al., 2022), and RoTTA (Yuan et al., 2023).

**Zero-shot** Zero-shot means that we use the base model, a pre-trained CLIP model, to be directly evaluated on the test data streams. The prompt applied to the text encoder is the commonly used template "a photo of a [CLASS]".

**ERM** In this baseline, the CLIP model is trained with the training sets of all domains. The textual prompt is trainable while the CLIP model itself is frozen, as in CoOp (Zhou et al., 2022b). The initialization of the prompt is the template "a photo of a [CLASS]". The training recipe is consistent with that in the prompt collection stage of our method. After training, the model with the trained prompt is evaluated on the test streams.

**TENT** TENT employs an entropy minimization loss to increase prediction confidence on test data, enabling adaptation during test time.

**SAR** SAR applies entropy minimization loss to filtered, reliable samples and further reduces the sharpness of the entropy.

**CoTTA** This method uses a teacher-student self-training framework, where the student model is continually trained with the data in streams and pseudo labels from the teacher model. The teacher model is updated by an exponential moving average of the student weights, and it produces the pseudo labels with test-time augmented input data. In the continual test-time adaptation setting, the method continually optimizes a source model in the streams of sequential target domains. In our experiments, for consistency with other baselines and our method, this method is evaluated starting from the ERM-trained model in streams.

**RMT** The method adopts a similar teacher-student self-training framework as CoTTA while further introducing a contrastive learning method with computed source prototypes and a source replay strategy. In our experiments, we also use the ERM-trained model as the initialization for this method. We use the training sets to compute the prototypes and conduct the source replay. Due to memory constraints, we do not include the evaluation of this method on the ImageNet-C dataset (1000 classes) because it needs to simultaneously keep two computational graphs for backpropagation for two losses associated with two different inputs.

**RoTTA** RoTTA is also based on a teacher-student self-training framework while it introduces a resampling approach to enhance stability and performance in online streams. We also evaluate the method using the ERM-trained base model.

## 4.4 IMPLEMENTATION DETAILS

In the prompt collection stage, we train the base model for 100 epochs on the training sets of each domain, respectively. Similar to Wang et al. (2024a), we collect one prompt sample per epoch during the last 80 epochs, resulting in a total of 80 prompt samples per domain. We use the SGD optimizer with a learning rate of 0.003, a momentum of 0.9, and a weight decay of 0.0003. To prevent overfitting, a cosine learning rate scheduler with a warmup period of 2 epochs is applied.

In the diffusion training stage, we use a denoising autoencoder for the latent space, following the architecture in Wang et al. (2024a). For the diffusion model, we adopt an architecture similar to

Table 2: Continual adaptation on test sets of 6 sequential domains in DomainNet. Continual TTA methods are evaluated using the ERM-trained model, with accuracy represented by the numbers. Some continual TTA methods only maintain the performance of the base model, similar to ERM, while our method demonstrates better deployment performance.

| Method | clip | info | paint | quick | real | sketch | Mean |
|---|---|---|---|---|---|---|---|
| Zero-shot | 71.0 | 47.6 | 66.2 | 13.9 | 83.7 | 63.5 | 57.7 |
| ERM | 75.3 | 55.6 | 72.3 | 25.5 | 85.9 | 68.0 | 63.8 |
| TENT | 75.4 ±0.01 | 53.6 ±0.02 | 69.5 ±0.04 | 1.8 ±0.05 | 84.4 ±0.02 | 50.4 ±1.80 | 55.9 ±0.30 |
| SAR | 75.0 ±0.03 | 53.5 ±0.09 | 69.8 ±0.07 | 11.7 ±0.84 | 85.0 ±0.17 | 65.1 ±0.40 | 60.0 ±0.19 |
| CoTTA | 75.3 ±0.03 | 55.4 ±0.02 | 72.2 ±0.06 | 23.4 ±0.58 | 85.4 ±0.28 | 67.2 ±0.37 | 63.2 ±0.22 |
| RMT | 74.7 ±0.18 | 55.0 ±0.15 | 70.8 ±0.35 | 20.0 ±1.76 | 85.9 ±0.16 | 67.6 ±0.41 | 62.4 ±0.48 |
| RoTTA | 75.4 ±0.05 | 55.5 ±0.05 | 72.1 ±0.08 | 22.6 ±1.24 | 85.5 ±0.22 | 67.5 ±0.30 | 63.1 ±0.30 |
| Ours | **79.6** ±0.06 | **58.9** ±0.23 | **75.8** ±0.69 | **30.1** ±0.14 | **87.7** ±0.48 | **71.5** ±0.50 | **67.3** ±0.29 |

Table 3: Continual adaptation on test sets from 15 sequential domains in ImageNet-C. Continual TTA methods are evaluated using the ERM-trained model. The numbers represent accuracy.

| Method | bri | gla | jpe | con | def | imp | mot | sno | zoo | fro | pix | gau | ela | sho | fog | Mean |
|---|---|---|---|---|---|---|---|---|---|---|---|---|---|---|---|---|
| Zero-shot | 53.0 | 15.1 | 32.1 | 21.6 | 23.1 | 14.6 | 24.1 | 28.1 | 22.0 | 28.5 | 32.3 | 15.2 | 13.2 | 15.7 | 38.1 | 25.1 |
| ERM | 57.6 | 19.9 | 36.7 | 26.7 | 27.7 | 20.2 | 29.4 | 33.4 | 26.8 | 33.4 | 38.4 | 20.3 | 18.6 | 20.2 | 43.2 | 30.2 |
| TENT | 46.7 ±0.04 | 5.5 ±0.23 | 0.6 ±0.01 | 0.2 ±0.02 | 0.1 ±0.00 | 0.2 ±0.01 | 0.1 ±0.00 | 0.1 ±0.00 | 0.1 ±0.01 | 0.2 ±0.02 | 0.1 ±0.00 | 0.1 ±0.00 | 0.1 ±0.00 | 0.1 ±0.00 | 0.1 ±0.01 | 3.6 ±0.01 |
| CoTTA | 46.1 ±0.04 | 12.9 ±0.03 | 28.2 ±0.06 | 19.7 ±0.07 | 19.9 ±0.10 | 12.7 ±0.07 | 20.4 ±0.10 | 23.7 ±0.10 | 18.0 ±0.21 | 23.8 ±0.23 | 26.7 ±0.34 | 12.5 ±0.16 | 10.1 ±0.12 | 12.6 ±0.39 | 31.4 ±0.19 | 21.3 ±0.13 |
| RoTTA | 46.2 ±0.03 | 13.2 ±0.18 | 28.9 ±0.28 | 20.5 ±0.45 | 20.9 ±0.41 | 13.8 ±0.47 | 20.9 ±0.22 | 24.3 ±0.26 | 18.6 ±0.14 | 23.8 ±0.25 | 26.7 ±0.35 | 12.7 ±0.15 | 10.4 ±0.06 | 12.7 ±0.40 | 30.0 ±0.90 | 21.6 ±0.07 |
| Ours | **60.8** ±0.00 | **21.0** ±0.00 | **38.4** ±0.05 | **27.9** ±0.05 | **28.3** ±0.00 | **20.9** ±0.05 | **30.5** ±0.05 | **35.4** ±0.04 | **28.3** ±0.00 | **34.6** ±0.05 | **39.4** ±0.05 | **20.4** ±0.00 | **22.0** ±0.04 | **20.8** ±0.05 | **45.5** ±0.00 | **31.6** ±0.01 |

Stable Diffusion (Rombach et al., 2022), but we scale down the model complexity considering the prompt data scale and add linear layers at both the beginning and end to fit the input and output to a 1D latent space. We use a linear Beta scheduler with $\beta_{start} = 0.0001$, $\beta_{end} = 0.02$, and 1000 steps. For this stage, we use the AdamW optimizer (Loshchilov & Hutter, 2019) with a weight decay of 2e-6 and a learning rate of 0.003. Both stages can be performed on a single GPU with 20GB of memory. The code will be available online.

## 4.5 RESULTS

**On sequential domains** Tables 2 and 3 present the results of continual adaptation on sequential domains for DomainNet and ImageNet, respectively. We perform multiple runs with five different random seeds for the evaluation. In Table 2, TENT and SAR degrade the performance of the ERM-trained model in the stream under the prompt-tuning paradigm. Regarding continual Test-Time Adaptation (TTA) methods, CoTTA and RoTTA retain stable performance, aligning with the ERM-trained model. Although these methods can improve performance from the source to the target domain by learning from test-time data of unseen domains, they do not further enhance performance on sequential domains seen by the ERM-trained model. Continual TTA methods focus on learning from unseen domains to prevent model degradation while overlooking performance gains in in-distribution deployment. In contrast, our method demonstrates better deployment performance than the ERM baseline by 3.6%. In Table 3, the continual TTA methods fail to maintain performance with the base model, while our generative expert prompts result in a 1.4% improvement over the ERM-trained prompt.

**On long-term recurring streams** Tables 4 and 5 present results of continual adaptation on long-term sequential and recurring domains for DomainNet and ImageNet, respectively. Here, the domain sequence recurs 15 times. Since each domain is repeated multiple times with different random seeds, we conduct the long-term evaluation only once. CoTTA and RoTTA lead to gradually degraded performance along the recurring episodes. RMT presents stable accuracy approaching that of the ERM-trained model, benefiting from the training prototype pulling. Meanwhile, our method showcases stability with the recurring domain sequence and performs better than the ERM baseline.

Table 4: Continual adaptation on a recurring 6-domain sequence in DomainNet, repeated 15 times. Continual TTA methods are evaluated using the ERM-trained model, with the numbers representing accuracy. While the following continual TTA methods show gradual degradation over the recurring sequences, our method demonstrates stability.

| Method | 1 | 2 | 3 | 4 | 5 | 6 | 7 | 8 | 9 | 10 | 11 | 12 | 13 | 14 | 15 | Mean |
|---|---|---|---|---|---|---|---|---|---|---|---|---|---|---|---|---|
| Zero-shot | 57.7 | 57.7 | 57.7 | 57.7 | 57.7 | 57.7 | 57.7 | 57.7 | 57.7 | 57.7 | 57.7 | 57.7 | 57.7 | 57.7 | 57.7 | 57.7 |
| ERM | 63.8 | 63.8 | 63.8 | 63.8 | 63.8 | 63.8 | 63.8 | 63.8 | 63.8 | 63.8 | 63.8 | 63.8 | 63.8 | 63.8 | 63.8 | 63.8 |
| CoTTA | 63.6 | 63.0 | 62.5 | 62.0 | 61.7 | 61.4 | 61.2 | 61.0 | 60.8 | 60.6 | 60.4 | 60.3 | 60.1 | 60.0 | 59.8 | 61.2 |
| RMT | 61.4 | 61.9 | 62.2 | 62.4 | 62.5 | 62.6 | 62.7 | 62.7 | 62.8 | 62.8 | 62.9 | 62.9 | 62.9 | 62.9 | 62.9 | 62.6 |
| RoTTA | 63.7 | 63.0 | 62.1 | 60.9 | 59.5 | 57.6 | 55.5 | 53.3 | 50.9 | 48.5 | 46.2 | 44.0 | 42.2 | 40.6 | 39.1 | 52.5 |
| Ours | **67.2** | **67.1** | **67.1** | **66.9** | **67.2** | **67.1** | **67.0** | **67.2** | **66.9** | **67.1** | **67.3** | **67.3** | **67.1** | **67.2** | **67.2** | **67.1** |

Table 5: Continual adaptation on a recurring 15-domain sequence in ImageNet-C. The sequence recurs 15 times. Continual TTA methods are evaluated using the ERM-trained model. The numbers represent accuracy.

| Method | 1 | 2 | 3 | 4 | 5 | 6 | 7 | 8 | 9 | 10 | 11 | 12 | 13 | 14 | 15 | Mean |
|---|---|---|---|---|---|---|---|---|---|---|---|---|---|---|---|---|
| Zero-shot | 25.1 | 25.1 | 25.1 | 25.1 | 25.1 | 25.1 | 25.1 | 25.1 | 25.1 | 25.1 | 25.1 | 25.1 | 25.1 | 25.1 | 25.1 | 25.1 |
| ERM | 30.2 | 30.2 | 30.2 | 30.2 | 30.2 | 30.2 | 30.2 | 30.2 | 30.2 | 30.2 | 30.2 | 30.2 | 30.2 | 30.2 | 30.2 | 30.2 |
| CoTTA | 21.5 | 21.3 | 20.7 | 20.0 | 19.1 | 18.3 | 17.5 | 16.8 | 16.1 | 15.4 | 14.8 | 14.2 | 13.7 | 13.2 | 12.7 | 17.0 |
| RoTTA | 21.6 | 21.4 | 19.9 | 18.6 | 17.7 | 16.9 | 16.3 | 15.7 | 15.2 | 14.7 | 14.3 | 13.9 | 13.6 | 13.2 | 12.9 | 16.4 |
| Ours | **31.6** | **31.6** | **31.5** | **31.6** | **31.6** | **31.6** | **31.6** | **31.6** | **31.6** | **31.6** | **31.6** | **31.6** | **31.6** | **31.6** | **31.6** | **31.6** |

## 4.6 ABLATION STUDIES

**Does the condition module perceive different domains?** The condition module is designed to perceive different domains so that it can produce conditions from images to guide the diffusion process in a domain-aware manner. To verify this, we visualize the output conditions of the condition module fed with test images from different domains. As shown in the t-SNE visualization in Figure 3a, the conditions from test images of different domains present a visually distinguishable distribution, which proves the domain-aware ability of the condition module.

**Are the image-conditioned generative prompts domain-specific?** The generative prompts conditioned on the incoming batch of images are expected to be domain-specific and tailored to the current domain. To verify this, we conduct an inter-domain-condition experiment. In this experiment, we test the generative prompts conditioned on images from one domain across the sequential domain streams. As shown in Figure 3b, each colored line represents a different domain of images that the prompt generation is conditioned on. The names surrounding the circles indicate the test domains. It can be observed that, for each test domain, the generative prompts conditioned on images from the same domain exhibit the best performance compared to those conditioned on images from other domains. This demonstrates that the generative prompts are domain-specific and well-suited for the encountered domain.

Table 6: Hypernetwork VS. Diffusion model. As a discriminative model, the hypernetwork baseline fails to model prompt distribution, leading to worse performance than the diffusion method.

| Method | clip | info | paint | quick | real | sketch | Mean |
|---|---|---|---|---|---|---|---|
| Hypernetwork | 65.7 | 46.6 | 59.9 | 12.8 | 77.2 | 60.1 | 53.7 |
| Diffprompt | **79.6** | **58.9** | **75.8** | **30.1** | **87.7** | **71.5** | **67.3** |

**What if a hypernetwork replaces the diffusion model for prompt generation?** We replace the diffusion model with a custom hypernetwork baseline. Specifically, the hypernetwork uses CLIP's image encoder as the backbone and a linear layer to generate the prompts. Following the similar training recipe described in Section 3.2, we present the results on DomainNet in Table 6. As a discriminative model, the hypernetwork baseline fails to model the prompt distribution from the collected prompts, leading to inferior performance compared to the diffusion-based method.

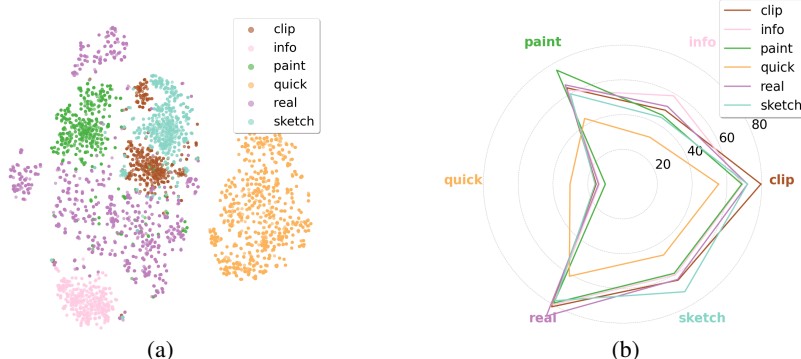

(a)                                   (b)

Figure 3: (a) Visualization of the conditions produced by the condition module. (b) Performance of inter-domain-conditioned generative prompts on sequential domain streams. Different colors in the legend indicate different domains of condition images.

Table 7: Comparison of GPU memory usage, computation cost, and model size. DiffPrompt demonstrates efficiency in both memory usage and computation cost.

| Method | Memory | Computation cost | Model size |
|---|---|---|---|
| TENT | 12.6 GB | 12.3 TFLOPs | 523.5 MB |
| CoTTA | 15.5 GB | 29.0 TFLOPs | 523.5 MB |
| RoTTA | 14.3 GB | 29.0 TFLOPs | 523.5 MB |
| RMT | 24.4 GB | 45.3 TFLOPs | 523.5 MB |
| DiffPrompt | 3.4 GB | 9.9 TFLOPs | 523.5 + 183.1 MB |

**Computation resources.** We compare the proposed method with the baselines in Table 7. We use PyTorch Profiler to record GPU memory usage, computation cost with a batch size of 64 on DomainNet, and compute the model size. Specifically, the GPU memory and computation cost account for operations during adaptation and inference at test time. As a result, DiffPrompt has lower memory and computation consumption because prompt generation does not involve gradient backpropagation, despite the 1000 denoising steps in the diffusion model's inference. In contrast, the optimizer-based baselines require calculating the gradient of each CLIP parameter, and their total cost would continue to increase as CLIP scales up. Additionally, a drawback of DiffPrompt is its model size, as it requires saving an extra diffusion model for prompt generation beyond the CLIP.

## 5    LIMITATION AND DISCUSSION

In this work, the setting does not include the domain generalization problem, where a model learns to generalize to an unseen domain from seen domains. The proposed prompt generation method learns a prompt-space distribution from the training sets of seen domains, without the aim of generalizing to the distribution of an unseen domain. Therefore, experiments show that generative prompts conditioned on images of an unseen domain only result in moderate performance similar to the zero-shot method. On the other hand, in Stable Diffusion (Rombach et al., 2022) for image generation, the diffusion model is trained with a wide range of conditions, while in our case, the model is trained with image conditions from only limited domains. This is a reason why this work does not primarily consider the generalization problem.

## 6    CONCLUSION

In this paper, we introduce Lifelong Continual Adaptation, which enables models to efficiently retrieve domain-specific knowledge when encountering sequential and recurring in-distribution data streams. For this realistic setting, we propose a novel prompt generation method that leverages a diffusion model to learn a prompt-space distribution for domains. During deployment, it generates domain-specific prompts conditioned on incoming images to adapt foundation models. We demonstrate that our generative prompts enhance model performance in practical data streams compared to baselines. Future work could explore integrating more diverse conditions into the prompt-space diffusion model training to improve generalization across unseen domains.

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
