# OpenReview forum: "Diffusion-based Prompt Generation for Lifelong Continual Adaptation"
_ICLR.cc/2025/Conference — ICLR 2025 Conference Withdrawn Submission_

### Official Review · Reviewer_s2MF · 2024-10-31

**Soundness:** 3
**Presentation:** 2
**Contribution:** 2
**Rating:** 3
**Confidence:** 4

**Summary:**

This paper introduces DiffPrompt, a method for lifelong continual adaptation of foundation models using diffusion-based prompt generation. The paper demonstrates the effectiveness of DiffPrompt in practical scenarios involving sequential and recurring domains, showing stable and efficient deployment.

**Strengths:**

1. This paper follows an interesting problem.
2. The proposed model enables stable and efficient deployment in practical scenarios involving sequential and recurring domains.

**Weaknesses:**

1.The authors need to further explain why DiffPrompt differs from existing prompt tuning techniques, and what key innovations make it suitable for long-term continual adaptation scenarios.
2.Could the authors clarify the impact on model performance when initialized with different prompts? Furthermore, how consistent are the experimental results across different initializations?
3.In the prompt collection stage, how sensitive is the performance of DiffPrompt to the number of prompt samples collected per domain and the total number of epochs used for training the base model?
4.The authors need to provide additional ablation studies to analyze the contribution of each component within the DiffPrompt framework. For example, what is the impact of removing the condition module or using a simpler prompt generation method?

**Questions:**

Please refer to weaknesses.

---

### Official Review · Reviewer_9W6n · 2024-11-02

**Soundness:** 2
**Presentation:** 3
**Contribution:** 2
**Rating:** 3
**Confidence:** 4

**Summary:**

The paper proposed a novel approach for Continual Test-Time Adaptation (CTTA). Traditional CTTA methods focus on handling sequential out-of-distribution data but fail to efficiently manage long-term and recurring in-distribution data streams. Moreover, existing methods may require significant computational resources and can exhibit instability when adapting to recurring domains over extended periods.

To overcome these limitations, a diffusion-based prompt generation method DiffPrompt is proposed. Rather than continually updating the foundation model’s parameters, DiffPrompt leverages a conditional diffusion model to generate domain-specific prompts, which enables the foundation model to adapt dynamically to the current domain via prompts. This method effectively sidesteps the costly and unstable parameter updates typical in traditional approaches. Experimental results demonstrate that DiffPrompt is effective.

**Strengths:**

- The paper is well-written and easy to follow. The motivation and the idea are clearly explained.
- Experiments of different settings are conducted and compared with SOTA methods. Ablation studies are conducted.

**Weaknesses:**

- My major concern is the new problem setting introduced in this paper.
  - The authors should show evidence that test-time adaption is needed for in-distribution data streams with sequential and recurring domains.
    - Methods from temporal domain adaptation or temporal domain generalization can be used, maybe they perform very well and no test-time adaptation is needed. These methods do not even require extra training of prompt generators, which is more efficient.
  - The model contains a foundation model and a prompt. Is there a baseline with the same foundation model but tested with no prompts?
- Could the authors provide an evaluation of the model's performance for the out-of-distribution (OOD) setting? The contribution is marginal if the model can work only for in-distribution (ID) data while not for OOD data. In real-world applications, prior knowledge of whether the test set is in-distribution (ID) or out-of-distribution (OOD) is typically unavailable.
- For the hypernetwork ablation study, what is the input to the hypernetwork? Is it domain embeddings or instance embeddings?

**Questions:**

Questions are asked in the weakness section.

---

### Official Review · Reviewer_nJK6 · 2024-11-04

**Soundness:** 3
**Presentation:** 3
**Contribution:** 3
**Rating:** 5
**Confidence:** 5

**Summary:**

In this paper, the authors introduce Lifelong Continual Adaptation, a method designed to enable models to efficiently retrieve domain-specific knowledge when processing sequential and recurring in-distribution data streams. To address this realistic setting, the authors propose a prompt generation technique that leverages a diffusion model to learn a prompt-space distribution for different domains. During deployment, this approach generates domain-specific prompts conditioned on incoming images, adapting foundation models accordingly. The authors demonstrate that these generative prompts enhance model performance in practical data streams compared to baselines.

**Strengths:**

1. The paper is well-written, with a clear and well-conveyed message.

2. The paper addresses the largely overlooked aspect of recurring in-distribution domain recurrence in continual adaptation methods, which is an insightful contribution.

3. Extensive experiments have been conducted, comparing the proposed method with state-of-the-art approaches in continual adaptation.

**Weaknesses:**

1. The authors claim that the setting of recurring in-distribution domains is more realistic; however, I disagree. In real-world scenarios, new out-of-distribution (OOD) domains are more likely to appear. Relying solely on this model could lead to failure in adapting to OOD domains, potentially posing risks in real-time applications (e.g., autonomous vehicles). I struggle to see a scenario where this setting is more applicable.

2. What impact would it have if the order of domain recurrence were altered? It seems the algorithm may not rely on the sequence of domain appearances.

3. A significant limitation of this setup is that introducing a new domain may require retraining each model from scratch, which is impractical.

4. For the ImageNet-C dataset, do the authors generate 15 noisy domains for training? If so, this approach uses significantly more data compared to methods like CoTTA, which rely only on clean data. This comparison may not be entirely fair.

5. There are also recent diffusion-based test-time adaptation methods, such as [A] and [B]. Comparisons with these approaches are needed for completeness.
[A] Turn Down the Noise: Leveraging Diffusion Models for Test-Time Adaptation via Pseudo-Label Ensembling
[B] Diffusion-TTA: Test-Time Adaptation of Discriminative Models via Generative Feedback

**Questions:**

See the weakness section.

---

### Note · Authors · 2024-11-13

I have read and agree with the venue's withdrawal policy on behalf of myself and my co-authors.